# Online Exercise Training Program for Brazilian Older Adults: Effects on Physical Fitness and Health-Related Variables of a Feasibility Study in Times of COVID-19

**DOI:** 10.3390/ijerph192114042

**Published:** 2022-10-28

**Authors:** Wagner Albo da Silva, Valéria Feijó Martins, Aline Nogueira Haas, Andréa Kruger Gonçalves

**Affiliations:** LaBiodin Biodynamics Laboratory, School of Physical Education, Physiotherapy and Dance, Federal University of Rio Grande do Sul, Porto Alegre 90010-150, Brazil

**Keywords:** older adult, internet-based intervention, exercise, feasibility studies, COVID-19

## Abstract

The COVID-19 pandemic brought negative consequences such as social isolation and limited access to health services, especially for older adults. The objective was to evaluate effects of an online exercise training program and physical fitness and health-related variables on Brazilian older adults during the COVID-19 pandemic and secondarily to assess the feasibility and application of an online program. A study was developed with twenty older adults who participated in a 9-month online exercise program. The physical fitness, depressive symptoms, concern about falling, and quality of life were assessed pre- and post-intervention. One-way repeated measures ANOVA and effect size was used. The feasibility was proven by the adherence to the program, in addition to the absence of identification of adverse effects. The results showed that physical fitness was improved (upper limb strength) or maintained (lower limb strength, lower and upper limb flexibility, cardiorespiratory fitness), as well as for most of the health-related variables (depressive symptoms, concern about falling, and quality of life domains). The study was developed in the first COVID-19 lockdown in Brazil, but positive and important results were obtained. This research supports the feasibility of the online exercise training program and provides a basis for an online exercise program for older adults.

## 1. Introduction

During the COVID-19 pandemic, older adults were encouraged to stay at home, which reduced the physical activity levels and increased sedentary behavior and risk of frailty, negatively affecting physical function and health-related variables [1,2]. Physical inactivity has been associated with cognitive decline, increased prevalence of chronic diseases, anxiety, and depression, and may also lead to a significant decline in functional capacity [3].

Physical exercise is considered a non-drug therapy for aging-related problems. Evidence suggests that multicomponent physical exercise programs are more suitable for older adults [3,4]. Improvements in cardiorespiratory fitness, muscle strength, flexibility, cognition, quality of life, balance and gait improvements, and reduction in the risk of falls have been highlighted [5]. International recommendations state that older adults should engage in moderate-intensity cardiorespiratory exercise for 150 to 300 min per week and in strength training at least two to three times per week [6,7]. According to these organizations, online physical activity is a safe way to maintain an active and healthy lifestyle in times of social distancing measures [6,7].

Digital technology has been used to mitigate the impacts of social distance measures and physical inactivity during the COVID-19 pandemic, remotely enabling social interaction and participation in physical exercise interventions [8]. The use of digital technology, digital social networks, and video conferencing platforms has contributed to increasing the feasibility of online physical training programs for older adults [8,9]. For these populations, online physical exercise is an excellent strategy to decrease the negative effects of physical inactivity and sedentary lifestyle, according to these last authors. The advantages of online physical activity include reducing financial costs (e.g., savings in commuting expenses), and increasing accessibility for a wider range of populations (e.g., those living in remote places, in unfavorable climate conditions, people with social restrictions or certain medical conditions) [10,11].

Investigations on the effects of online structured physical exercise programs on physical fitness and health-related variables of older adults are still incipient. The COVID-19 pandemic online physical activity interventions were shown to be feasible and acceptable strategies to help people stay healthy [2,6,7,8]. However, investigations on the effects of online physical activity programs on physical fitness and health-related variables of older adults are still incipient. Most are descriptive studies, and few have an intervention design [1,3,10,11]. Evidence suggests that at-home physical exercises seem to be effective in improving physical fitness, but at least a minimal level of supervision is recommended [10].

Given the unique context afforded by the COVID-19 pandemic as well as the beneficial proporciones for exercise programs, this study was to evaluate the effects of an online exercise training program and physical fitness and health-related variables on Brazilian older adults during the COVID-19 pandemic, and secondarily to assess the feasibility and application of an online program. It is believed the adopted intervention model and its periodization may be beneficial to older adults in general, not solely in times of social restrictions.

## 2. Materials and Methods

### 2.1. Study Design

This is a 9-month quasi-experimental study. The restrictions imposed by the COVID-19 pandemic in 2020, made it impossible for the inclusion of a control group, given the social restrictions in place during the study period, which is why the decision to proceed with characteristics of a feasibility study.

The pre-intervention evaluations were conducted in-person in December 2019 (before the pandemic), and the post-intervention, also in-person, at the beginning of December 2020, when social distance restrictions were eased in Brazil. The study occurred during the first wave of the COVID-19 pandemic and in Brazil the mortality and morbidity rates were high, especially in the older population. The vaccine did not exist and the health conditions were even worse due to social distancing that restricted access to health promotion actions.

### 2.2. Participants

The participants were part of the extension project CELARI (Centro de Estudos de Lazer e Atividade Física do Idoso) from the Federal University of Rio Grande do Sul, Brazil. Inclusion criteria consisted of independent, community dwelling, older adults >60 years old who were approved to practice physical exercise by a physician, participated in the in-person pre-intervention evaluation in December 2019, attended at least 70% of classes throughout the year previous, have access to Facebook and be part of the project’s private group on this social network. Exclusion criteria consisted of medical conditions preventing physical activity including any question orthopedic, neurological, or cardiovascular, frequency less than 70% in program, and not having accessibility to carry out the final in-person assessment according to the safety protocols established for the COVID-19 pandemic in December 2020.

The outline of the trial and the sample distribution process are shown in Figure 1. Sample size was calculated using the G*Power Version 3.1 software (significance level 5%, test power 80%, effect size 0.58) [12,13], resulting in a sample of 18 participants. Considering the possibility of drop outs, the sample size was increased by 10%, totaling 20 participants. Thus, the final sample consisted of 20 Brazilian older adults, 18 women and 2 men with a mean age of 70.5 ± 4.4 years (Figure 1).

### 2.3. Intervention

The online and live exercise training program was provided three times a week on alternate days through a private group on Facebook. The technological tools selected were Facebook group pages pre-existing and instructors used WhatsApp group chats to deliver updates and reminders. In the live session, two team members acted, one of which was to explain and develop the exercises with the count or time of each one, while the second was the assistant, assuming technical challenges arose during the live-streamed. Participants had the choice of asking for help in the group chat or in a private message.

Figure 2 presents the construction of the study from the context of the COVID-19 pandemic. Older persons evaluated at the end of 2019 were invited in March 2020 to participate in online physical training. An innovation for the majority and that was a possible strategy in the first wave, in which the cases of contamination and deaths increased substantially and with severe sanitary restrictions, as well as social ones. In a quick organization, a training methodology was structured with scarce and almost non-existent references, regarding the older public. Our online training started in March, over 9 months with three weekly sessions, ending in December 2020 with the final and face-to-face assessment, to enable comparison with the year 2019. Until the time of structuring the methodology, there were no investigations with these characteristics: online intervention with physical exercise for elderly people for 9 months. Thus, the online exercise training program was carried out for nine months, with one month of adaptation to the training and the objectives were updated every two months, composing 4 blocks of 8 weeks with three weekly sessions, as shown in Figure 3. The weekly sessions were developed using the same logic with the same objective as the corresponding block, but with different exercises in each session. The sessions were organized in (1) Warm-up of 5 min with joint mobility, dynamic warm-up and gait variations; (2) main part of 45–50 min with two repetitions of the same circuit of specific exercises according to the objective of the periodization; and (3) cool-down of 5–10 min with flexibility, breathing, and relaxation exercises. The materials used for training such as chairs, cushions, water bottles, milk cartons, food packages, broom handles, books, blankets (mat replacement), etc., were easily available at home. The participants were informed what materials would be used in each class.

### 2.4. Instruments

The participants completed an anamnesis providing sociodemographic data (age, gender, marital status, income, level of education, and living arrangement) and health characteristics.

Physical fitness was assessed using the Senior Fitness Test [14], a battery composed of six physical tests measuring lower limb strength (30-s chair-stand test, measured in number of repetitions completed), upper limb strength (arm-curl test, measured in number of repetitions completed), lower limb flexibility (chair-sit-and-reach test, measured in centimeters), upper limb flexibility (back scratch test, measured in centimeters), balance and agility (8-foot up-and-go test, measured in seconds), and cardiorespiratory fitness (2-min step test, measured in repetitions) [14]. The results measured in repetitions or centimeters are directly proportional to the best performance, while the results in time (seconds) are inversely proportional.

The Geriatric Depression Scale-Short Form (GDS-15) was used to assess depressive symptoms. It consists of 15 statements that can be answered positively or negatively. The sum of scores indicates the state of depressive symptomatology: severe depression (over 10 points), mild depression (6–10 points), and no depressive symptoms (less than 6 points) [15].

The Falls Efficacy Scale-International Brazil (FES-Brazil) [16] was used to assess concern about falls. Scores range from 16 to 64, with the lowest value indicating no concern about falls and the highest value representing extreme concern. The score is classified as not associated with falls (16 to 22 points), associated with sporadic falls (23 to 30 points), and associated with recurrent falls (≥31 points).

The Medical Outcomes Study 36-item Short-Form Health Survey (MOS SF-36) [17] was used to analyze health-related quality of life. The Medical Outcomes Study 36-item Short-Form Health Survey (MOS SF-36) is composed of 36 items grouped into eight domains: SF-36/physical functioning, SF-36/role-physical, SF-36/bodily pain, SF-36/general health, SF-36/vitality, SF-36/social functioning, SF-36/role-emotional, SF-36/mental health. Raw scores, ranging from 0 to 100, were obtained for each domain, with higher scores representing higher perceptions of health-related quality of life.

### 2.5. Statistical Analysis

Sociodemographic and health condition variables were analyzed using descriptive statistics to characterize the sample. The Shapiro–Wilk test was used to determine data distribution normality. One-way repeated measures ANOVA was performed (*p* ≤ 0.05) to determine the effect of intervention on measures of physical fitness and quality of life-related variables. F-statistic, eta-squared, and *p*-values are reported in text. The effect size was calculated with the values of the mean of the pre-test and the post-test. These score changes and their standard deviations were used to calculate the effect sizes according to Cohen’s *d* formula: *d* = M1 − M2 / SDpooled, where M1 is the pre-test, M2 is the post-test, and SDpooled is the weighted average of the standard deviation of pre- and post-test results [18]. Cohen’s *d* was used to predict effects interpreted as insignificant (≤0.19), small (0.20–0.49), medium (0.50–0.79), and large (≥0.80) (Cohen, 1992). The statistical tests were performed in SPSS^®^ version 23.0 (IBM Corporation, Armonk, NY, USA).

## 3. Results

### 3.1. Sample Characteristic

Twenty participants completed the training with more than 70% frequency. They were majority of the female (*n* = 18, 90%); married (*n* = 10, 50%); living with their spouse (*n* = 10, 50%); had 8 years or more of education (*n* = 18, 90%); with a family income of 4 to 6 minimum wages (*n* = 6, 30%) or more than 10 minimum wages (*n* = 6, 30%). The participants self-reported the following health conditions: high blood pressure (*n* = 11, 55%); diabetes (*n* = 1, 1%); osteoporosis (*n* = 2, 10%); arthritis (*n* = 4, 20%); arthrosis (*n* = 10, 50%), heart disease (*n* = 1, 5%), one fall (*n* = 4, 20%), and two or more falls (*n* = 1, 5%).

### 3.2. Physical Fitness Outcomes

Table 1 shows a comparison of the pre- and post-intervention physical fitness results. The repeated measures ANOVA suggested that there was statistically a main effect from the time (F (1.000,19.000) = 5.922; *p* = 0.025) and variable (F (2.737,52.000) = 516.151; *p* < 0.001) condition. The test was not significant for interaction time × variable (F (2.278,43.286) = 0.838; *p* = 0.453). The online exercise training program improved upper limb strength (*p* = 0.006), with a medium effect size (*d* = 0.59). The lower limb strength (*p* = 0.150), lower limb flexibility (*p* = 0.764), upper limb flexibility (*p* = 0.289), and cardiorespiratory fitness (*p* = 0.180) were unchanged and effect size insignificant or small. However, there was a significant deterioration in balance/agility (*p* < 0.001), with a large effect size (*d* = 1.16).

Differences between pre- and post-test physical fitness variables (delta, see Table 1) indicated an improvement of 1.2 repetitions for lower limb strength, 2.7 repetitions for upper limb strength, 0.5 cm for lower limb flexibility, 1.4 cm for upper limb flexibility, and 4.4 repetitions for cardiorespiratory fitness. The increase in repetitions or centimeters represents an improvement in the performance of the sample, with the exception of the time when the logic is inverse (less time, better result). Balance and agility showed a reduction of 0.64 s, that is, the time required to complete the balance/agility test increased.

### 3.3. Health-Related Variables

The pre- and post-intervention results for health-related variables can be seen in Table 1. The repeated measures ANOVA suggested that there was statistically main effect from the time (F (1.000,19.000) = 14.066; *p* = 0.001), variable (F (4.045,76.853) = 78.315; *p* < 0.001) condition, and for interaction time × variable (F (2.685,51.015) = 4.369; *p* = 0.010). The results of the interaction showed that the training did not overcome the effects on quality of life provided by the social isolation provided by COVID-19, that is, the SF-36/physical functioning (*p* = 0.001), SF-36/role-physical (*p* = 0.011), and SF-36/role-emotional (*p* = 0.018) domains showed a statistically significant decline. The SF-36 domains have a maximum score of 100 points, and most domains, in both evaluation periods, achieved satisfactory values, with scores greater than 70 points, showing that participants have a good quality of life, however the pandemic reduced these scores. Depressive symptomatology scores (*p* = 0.120), and concern about falls (*p* = 0.530) were unchanged. As for depressive symptomatology, it is noteworthy that mean scores increased in 2020 compared to 2019, indicating absence of depressive symptoms, but differences between years were not significant. The concern about falls showed a similar behavior, with mean values increasing in 2020, indicating a sporadic fear of falling.

## 4. Discussion

This study sought to evaluate application and effects of an online exercise training program and physical fitness and health-related variables on Brazilian older adults, providing evidence about a multicomponent physical training. The results revealed that the program is feasible to be carried out at home, in moments of distancing, by the older adult. Moreover, it was effective in improving the upper limb strength and maintaining the physical fitness variables (lower limb strength, lower limb flexibility, upper limb flexibility, and cardiorespiratory fitness), except the balance/agility. Furthermore, it resulted in the maintenance of health-related variables (GDS, FES, and SF-36 domains SF-36/bodily pain, SF-36/general health, SF-36/vitality, SF-36/social functioning, and SF-36/mental health), except the three SF-36 domains (SF-36/physical functioning, SF-36/role-physical, and SF-36/role-emotional).

Staying physically active with online exercise is a beneficial strategy to mitigate the effects of physical inactivity and sedentary lifestyle [2,9,13], and our study corroborates with this affirmation. Literature reports reinforce the need for physical exercise for older adults to maintain/improve physical fitness, physical function, and psychosocial function, thereby contributing to an independent, autonomous life [1,3,4,5].

In a randomized controlled trial with older adults that evaluated the effects of strength training performed at home without online supervision, it was possible to observe a significant improvement in lower limb strength in the experimental group compared with the control [19]. A study carried out between June and July 2021 evaluated the effects of a remote home-based exercise program on mental state, balance, and physical function and prevention of falls in the older adult [20]. The fall prevention exercise program improved all study variables, but pre-study exercise was not indicated.

These differences are likely due to the profile of the sample. Whereas participants of these studies were sedentary, our sample comprised physically active individuals. Physical training has the overload principle which indicates that “the more you train, the less trainable”. Our sample already had this pattern of behavior and the purpose of the intervention was for them to maintain what they had already acquired in face-to-face training.

By contrast, in our study, we found a significant improvement in upper limb strength and maintenance of lower limb strength. One explanation for such differences is related to the possibility of greater involvement in domestic activities (such as sweeping the floor, hanging clothes, and carrying shopping bags, among others) during the lockdown period, which might have led to greater use of upper body muscle groups.

Similar to our results, the benefits of the in-person multicomponent exercises for the older adult population were investigated, before the pandemic, showing improvement or maintenance on the physical fitness components [5,12,21]. The benefits of multicomponent exercises for the older adult population were investigated, showing that the training program with exercises of muscle strength, endurance, balance, and flexibility, presented notable improvement of functional skills and quality of life [5]. Moreover, the application of a multicomponent physical exercise protocol for older adults, face-to-face indicated significant improvements in muscle strength, flexibility, balance, and agility with the Senior Fitness Test [21]. A longitudinal, 5-year investigation of multicomponent training with older adults reported improvement of cardiorespiratory fitness and muscle strength of the lower limbs, as well as maintenance of balance and flexibility [12].

The results of the three previous studies indicated improvement or maintenance of physical fitness components, but interventions were not performed in the online format and were before COVID-19 pandemic. As far as we know, we did not find studies with older adults participating in a multi-component physical program during the pandemic.

Analysis of health-related quality of life revealed maintenance in several domains with scores closer to or higher than 70 out of 100, indicating a good quality of life [22,23] and a reduction in some domains (SF-36/physical functioning, SF-36/role-physical, and SF-36/role-emotional). Similar results were found in the older adults enrolled in an educational health program, before and during the COVID-19 pandemic, showing significant decline in the SF-36/physical functioning, SF-36/social functioning, and SF-36/mental health [24]. Previous studies about the older adult’s quality of life revealed that some domains are more responsive to physical exercise than others [22,23,24,25]. Our results corroborate with this affirmation, even though the intervention was carried out in a different format (online and live).

Two of the three domains that worsened are related to the influence of physical and emotional health on the performance of daily activities. The SF-36/physical functioning domain assesses the presence or extent of the limitation of physical capacity, whereas SF-36/role-physical and SF-36/role-emotional assess the difficulty of performing daily activities on an ordinary day associated with physical and emotional aspects [17]. The older adults of the current sample can be characterized as independent, and the restriction of social life became a cause for concern, as it could impair their functionality.

Regarding depressive symptoms and concern about falls, no changes were observed. The online exercise training program was able to maintain these variables in the atypical period of social distancing, representing a positive result. Studies have shown that older adults become more depressive or exhibit depressive symptoms [26,27] and increase their concern about falls as a result of staying at home due to the COVID-19 pandemic, which is the main place where falls occur, enhancing their susceptibility [28]. Engaging predominantly in domestic activities can decrease physical activity levels, resulting in a decline in functional capacity and, consequently, in an increase in the risk of falls [29]. Some studies showed association between physical activity levels and reduced depressive symptomatology in older adults who were physically active during social distancing via in-person training, showing that the ones who managed to maintain or improve their physical activity level did not exhibit new depressive symptoms [20,26,27,28,29]. These findings are similar to our results and provide evidence of the protective effect of physical activity on depressive symptoms.

The home-based exercise program generates positive results with the group of older adults in the midst of COVID-19. To the best of our knowledge, this is the first study investigating an applicable home-based training strategy for the older adults during the COVID-19 pandemic, and the effect of a long-term training (9-months). In this context of long stay-at-home restrictions, it seems that this intervention is an applicable strategy for the older adults [2,6,7,8].

Remote intervention or online or at a distance or Internet-based or e-health (defined by WHO as the use of information and communication technologies in health), in addition to a more recently used term remote delivery of health care and research, denote a commonplace that will be increasingly feasible. However, it is necessary to be attentive to economic and educational issues so that it does not become another form of social inequality [10].

We highlighted the relevance of our study results, including the maintenance of some of the physical fitness and health-related variables during 9-months of a periodized online exercise training program. Our investigation, carried out during the first year of the COVID-19 pandemic, sought an alternative to in-person interventions. The online format, a new mode of interaction that will be more present in the daily lives of the older population, afforded positive results for the participants after a safe and effective intervention.

There are some obstacles to obtaining the benefits of physical exercise for physical fitness and health variables when training at home. Lack of specific equipment or materials for physical training is one of these challenges. The difference between the results of face-to-face versus online interventions seems to be directly associated with the environment. Before the pandemic, individuals performed physical exercise in places suitable for this practice (as was the case with our sample), with the presence of teachers and specific equipment, at home the training intensity also differed. With social distancing, however, physical exercise began to be carried out in the domestic environment, which is not ideal for this purpose.

For the online format, bodyweight exercises, such as free squats, step-ups, and wall push-ups, are recommended. The use of adapted materials such as broom handles, water bottles, and food packages is indicated to vary and add load to exercises. Cardiorespiratory exercises such as walking in place and moving indoors are a good option, as are balance and agility exercises such as walking on a line on the floor, one leg stand, and walking over obstacles [2]. Unstable bases can be replaced by pillows, towels, and blankets, allowing one to meet one of the principles of balance training. Despite all these adaptations, the control of intensity in strength and cardiorespiratory fitness exercises is very limited in the home environment. Attention is drawn to the lack of adequate technique and correction of movements during the execution of exercises, which may negatively interfere with the effects of training [9,10].

Regarding the limitations of the study, we pointed out the difficulty in applying the assessment instruments in-person in the post test, because of the social distancing measures, resulting in a quasi-experimental study. Another limitation would be related to communication with the older adults, as contacts were made via social networks and message groups to encourage participation. The older adults in the study were invited to continue participating in the online training as an extension project, and most persisted. This is an important ethical issue because it is necessary to continue contact with the sample groups, either via continuity of intervention or scheduled return visit to verify the situation of the people.

## 5. Conclusions

The current study’s findings provide data on the efficacy of online exercise training in older adults. By investigating and promoting physical fitness and health-related variables in older adults, the online intervention was able to improve upper limb strength and maintain lower limb strength, upper and lower limb flexibility, and cardiorespiratory fitness. However, training was not effective for balance and agility, which declined over time. Among health-related variables, concern about falls, depressive symptomatology, and quality of life domains (bodily pain, general health, vitality, social functioning, and mental health), it was possible to observe a maintenance in the results, except for physical functioning, role-physical and role-emotional of quality of life showed a decline. Although the mean values of this variable indicated a good quality of life.

Our results are promising and reinforce the need for new studies addressing online exercise training programs for the older adult community. This type of program could be a reasonable option in case of inability to carry out face-to-face programs. Thus, as this study was developed in the first COVID-19 lockdown in Brazil, positive and important results were obtained for the sample.

Despite the limitation of not having similar studies to corroborate or compare our results, we can affirm that online physical training had a protective effect on most of the study variables in the COVID-19 pandemic, despite the negative impacts on health; especially for the older population.

## Figures and Tables

**Figure 1 ijerph-19-14042-f001:**
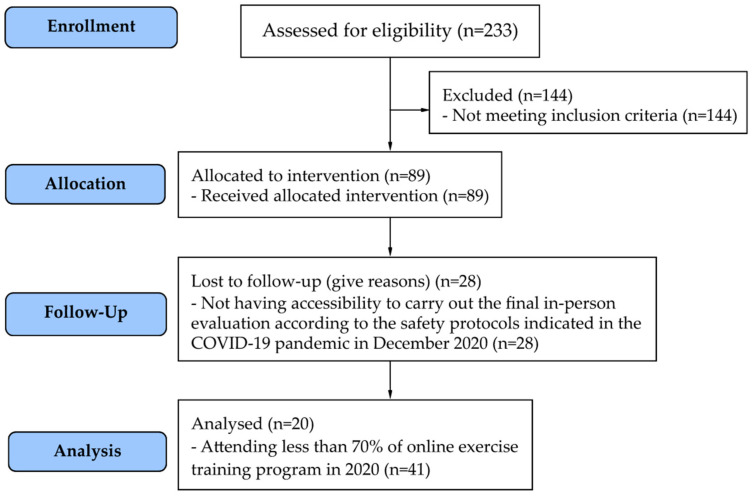
Sample distribution flow chart according to the inclusion and exclusion criteria.

**Figure 2 ijerph-19-14042-f002:**
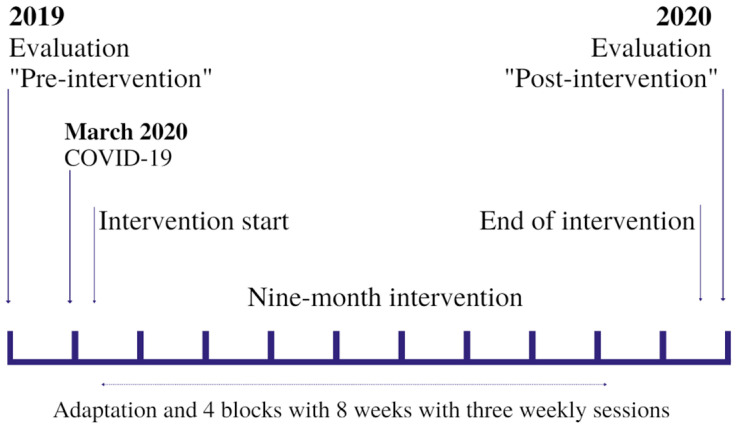
Study timeline with emphasis on evaluation moments.

**Figure 3 ijerph-19-14042-f003:**
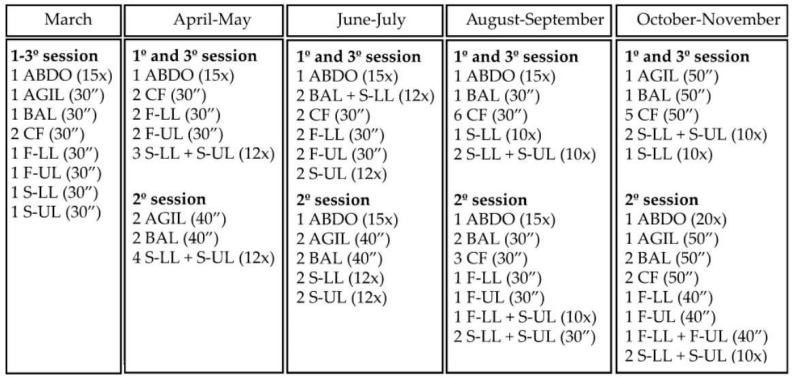
Periodization of the online physical training program and description of the amount of exercises of each component of physical fitness, for each session. Note. ABDO: abdomen/core, AGIL: agility; BAL: balance; CF: cardiorespiratory fitness; F-LL, lower limb flexibility; F-UL, upper limb flexibility; S-LL: lower limb strength; S-UL: upper limb strength; x: repetitions; ″: seconds. Microcycles in each macrocycle have the same objectives, microcycles 1 and 2 were performed on Monday and Friday (with different exercises in each session and same volume), microcycle 3 occurred on Wednesday (with different exercises and volume of microcycles 1 and 2).

**Table 1 ijerph-19-14042-t001:** Mean, standard deviation, and effect size of variables of physical fitness and quality of life of older adults before and after an online exercise training program.

Variables		2019	2020	*p*	*d*	∆
		Mean ± SD	Mean ± SD			
Physical Fitness	Lower limb strength (rep)	18.3 ± 5.1	19.6 ± 5.4	0.150	0.24	1.25
	Upper limb strength (rep)	20.1 ± 3.9	22.8 ± 5.2	0.001 *	0.59	2.75
	Lower limb flexibility (cm)	−0.7 ± 10.7	−0.1 ± 12.7	0.760	0.05	0.55
	Upper limb flexibility (cm)	−4.6 ± 9.6	−3.2 ± 10.6	0.290	0.14	1.46
	Balance and agility (s)	4.3 ± 0.5	4.9 ± 0.5	<0.001 *	1.16	0.64
	Cardiorespiratory fitness (rep)	95.2 ± 17.5	99.6 ± 13.9	0.180	0.28	4.40
Quality of life-related	Geriatric Depression Scale (pts)	1.8 ± 2.2	2.5 ± 1.8	0.120	−0.31	0.65
	Falls Efficacy Scale (pts)	22.6 ± 5.8	23.2 ± 4.9	0.530	−0.12	0.65
	SF-36/physical functioning (pts)	87.0 ± 16.9	71.4 ± 23.4	<0.001 *	0.76	−15.60
	SF-36/role-physical (pts)	93.7 ± 13.7	69.1 ± 40.5	0.001 *	0.81	−24.65
	SF-36/bodily pain (pts)	67.0 ± 20.2	69.2 ± 22.8	0.642	−0.09	2.15
	SF-36/general health (pts)	88.2 ± 37.1	71.9 ± 16.4	0.058	0.55	−16.25
	SF-36/vitality (pts)	77.7 ± 17.5	77.0 ± 18.2	0.742	0.03	−0.75
	SF-36/social functioning (pts)	88.1 ± 16.9	88.6 ± 19.4	0.878	−0.02	0.50
	SF-36/role-emotional (pts)	86.5 ± 27.5	72.1 ± 36.3	0.018 *	0.44	−14.46
	SF-36/mental health (pts)	83.4 ± 16.4	82.0 ± 18.1	0.551	0.08	−1.40

Note. SD: standard deviation; *: *p* < 0.050; *d*: effect size; Δ: mean difference between 2019 and 2020; rep: repetitions, cm: centimeters; s: seconds; pts: points.

## Data Availability

The datasets generated during and/or analyzed during the current study are available from the corresponding author on reasonable request.

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
