# Peer review of "Online Exercise Training Program for Brazilian Older Adults: Effects on Physical Fitness and Health-Related Variables of a Feasibility Study in Times of COVID-19"

_ijerph, 2022, doi:10.3390/ijerph192114042_

Round 1

Reviewer 1 Report

This paper investigated the relationship between physical fitness and physical activity among the elderly during the COVID-19 pandemic. Although I am not very familiar with this area, I have the following concerns:

1. The typesetting of the article needs to be improved. For example, there is an obvious blank on the third page.

2. The paper needs a more detailed flow chart to describe the innovativeness of the proposed method.

3. Picture 1 is a little too simple, and the description should go below the picture (other pictures have this problem too).

4. More content should be added to the method section, such as the proposed novel statistical method and the difference from previous methods.

5. The table in Section 3.2 requires a label.

Author Response

Dear Revisor 1 of IJERPH,

We would like to thank you for the opportunity, and respond to your comments. The comments made us improve the text and hope that all concerns have been properly addressed in our responses below. Our point-by-point responses are below in bold and marked in the text.

Revisor 1:

This paper investigated the relationship between physical fitness and physical activity among the elderly during the COVID-19 pandemic. Although I am not very familiar with this area, I have the following concerns:

  1. The typesetting of the article needs to be improved. For example, there is an obvious blank on the third page.

RESPONSE: Thank you, we apologize for the lack of care with the spaces that the figure needs. We reviewed all the text and adjusted the typography.

  1. The paper needs a more detailed flow chart to describe the innovativeness of the proposed method.

RESPONSE: In fact, the methodology could be better elucidated with a figure that highlights the innovativeness of our method. Thus, a flowchart was prepared followed by an explanation in the text.

“Figure 2 presents the construction of the study from the context of the COVID-19 pandemic. Older persons evaluated at the end of 2019 were invited in March 2020 to participate in online physical training. An innovation for the majority and that was a possible strategy in the first wave, in which the cases of contamination and deaths increased substantially and with severe sanitary restrictions, as well as social ones. In a quick organization, a training methodology was structured with scarce and almost non-existent references, regarding the older public. Our online training started in March, over 9 months with three weekly sessions, ending in December 2020 with the final and face-to-face assessment, to enable comparison with the year 2019. Until the time of structuring the methodology, there were no investigations with these characteristics: online intervention with physical exercise for elderly people for 9 months.” (line xx-xx)

  1. Picture 1 is a little too simple, and the description should go below the picture (other pictures have this problem too).

RESPONSE: Adjusted.

  1. More content should be added to the method section, such as the proposed novel statistical method and the difference from previous methods.

RESPONSE: Thanks, we've revised the text and included my information about the method.

“One-way repeated measures ANOVA was performed (p ≤ 0.05) to determine the effect of intervention on measures of physical fitness and quality of life-related variables. F-statistic, eta-squared, and P-values are reported in text.” (line xx-xx)

  1. The table in Section 3.2 requires a label.

RESPONSE: That's right, we've made the adjustments.

Reviewer 2 Report

Thank you for the opportunity to review this well-written study

Title: Please remove the word feasibility. I do not think you have performed a feasibility study since you do not use the proper framework and the proper steps. You do not cover strengths and weaknesses, marketing and economics. I think you can call your study an observational cohort study and use the STROBE framework. Please remove the word feasibility throughout the manuscript. 

Abstact: No comments

Introduction: No comments

Methods: Please add what time-points the instruments were used

Results: How did you measure upper and lower limb strength? Was it measured by the older persons themselves or by you? Plese explain.

line 223. Was your data normally distributed? I do not think so if you only have 20 participants. Please perform the calculations and correct if necessary. There is no number on the table. Please add one. What is it you are measuring in the table, kilograms or???

Discussion: Please add informations about the limitations of your study design. E.g. If you did not contact your participants several times before training, do you think they would perform the training? What is happening now with the participants´ training? Do you think they will continue training now the study has ended and you are not there to support them? Please elaborate.

References: I think you have too many references. 25-30 references are preferable.

Author Response

Dear Revisor 2 of IJERPH,

We would like to thank you for the opportunity, and respond to your comments. The comments made us improve the text and hope that all concerns have been properly addressed in our responses below. Our point-by-point responses are below in bold and marked in the text.

Revisor 2:

Thank you for the opportunity to review this well-written study.

Title: Please remove the word feasibility. I do not think you have performed a feasibility study since you do not use the proper framework and the proper steps. You do not cover strengths and weaknesses, marketing and economics. I think you can call your study an observational cohort study and use the STROBE framework.

Please remove the word feasibility throughout the manuscript.

RESPONSE: We agreed on the feasibility study and characterized it as a quasi-experimental study. All mention of the feasibility study has been deleted from the text. We appreciate the indication of the STROBE framework, but it is not consistent with what was accomplished.

Abstract: No comments

Introduction: No comments

Methods: Please add what time-points the instruments were used

RESPONSE: We appreciate the observation, but this information is in the 'study design' item: "The pre-intervention evaluations were conducted in-person in December 2019 (before the pandemic), and the post-intervention, also in-person, at the beginning of December 2020, when social distance restrictions were eased in Brazil." A sentence has been included at the end of the 'instruments' item:

"The instruments were collected before and after the intervention.” (line xx-xx).

Results: How did you measure upper and lower limb strength? Was it measured by the older persons themselves or by you? Please explain.

RESPONSE: the physical fitness tests were performed according to the Senior fitness test, proposed and validated by Rikli and Jones (1999). In which an evaluator performs all measurements or repetition counts, while the elderly person performs the action. We include this information in the text:

“Physical fitness was assessed using the Senior Fitness Test [24], where an evaluator is responsible for measuring and counting, while the subject performs the action. It should be noted here that all sanitary measures were taken during the post-intervention evaluation. This battery, […]” (line xx-xx).

line 223. Was your data normally distributed? I do not think so if you only have 20 participants. Please perform the calculations and correct if necessary.

RESPONSE: The normality calculation was revised and we confirmed that the sample has a normal distribution for the variables investigated.

There is no number on the table. Please add one. What is it you are measuring in the table, kilograms or???

RESPONSE: Thanks, sorry for the lack of attention, it has been adjusted. In relation to the measurement, it is described in the sequence of tests in the methods, we agree that the information is far from the table and we have included this information in the table (rep: repetitions, cm: centimeter; s: seconds; pts: points).

Discussion: Please add informations about the limitations of your study design. E.g. If you did not contact your participants several times before training, do you think they would perform the training? What is happening now with the participants´ training? Do you think they will continue training now the study has ended and you are not there to support them? Please elaborate.

RESPONSE: The limitations indicated are interesting and have been inserted into the text. Online training was a strategy unknown to most of the public, especially the older. However, in the COVID-19 pandemic, contact via videoconferencing and/or social media has become the main form of communication. In the case of our study, contacts were made via social networks and message groups to encourage participation. It is important to show this audience that their participation is possible, even if they have difficulties with digital technology. The older adults in the study were invited to continue participating in the online training as an extension project, and most persisted. This is an important ethical issue because it is necessary to continue contact with the sample groups, either via continuity of intervention or scheduled return visit to verify the situation of the people.

References: I think you have too many references. 25-30 references are preferable.

RESPONSE: Thanks, for the observation, we really exaggerated the number of references, we managed to make several adjustments and reduce it to 29 references.

Round 2

Reviewer 1 Report

The authors have addressed my previous concerns, but I would recommend adding more technical content to the methods section to enhance the article's creativity.

Author Response

Thanks for your return. As for the technical content, in the Methods section, we have included an explanation of the timeline of the study and intervention. We adjusted the statistical analysis section and some points of the instruments. Please review these changes as requested.

We include these informations in the text regarding who made the assessment, timeline and statistics:

"The instruments were collected before and after the intervention.”

“Physical fitness was assessed using the Senior Fitness Test [24], where an evaluator is responsible for measuring and counting, while the subject performs the action. It should be noted here that all sanitary measures were taken during the post-intervention evaluation. This battery, […]”.

“Figure 2. Study timeline with emphasis on evaluation moments.

Figure 2 presents the construction of the study from the context of the COVID-19 pandemic. Older persons evaluated at the end of 2019 were invited in March 2020 to participate in online physical training. An innovation for the majority and that was a possible strategy in the first wave, in which the cases of contamination and deaths increased substantially and with severe sanitary restrictions, as well as social ones. In a quick organization, a training methodology was structured with scarce and almost non-existent references, regarding the older public. Our online training started in March, over 9 months with three weekly sessions, ending in December 2020 with the final and face-to-face assessment, to enable comparison with the year 2019. Until the time of structuring the methodology, there were no investigations with these characteristics: online intervention with physical exercise for elderly people for 9 months.”

One-way repeated measures ANOVA was performed (p ≤ 0.05) to determine the effect of intervention on measures of physical fitness and quality of life-related variables. F-statistic, eta-squared, and P-values are reported in text..
